# Nanoscale geochemical and geomechanical characterization of organic matter in shale

Jing Yang[1], Javin Hatcherian[2], Paul C. Hackley[2] & Andrew E. Pomerantz[1]

Solid organic matter (OM) plays an essential role in the generation, migration, storage, and production of hydrocarbons from economically important shale rock formations. Electron microscopy images have documented spatial heterogeneity in the porosity of OM at nanoscale, and bulk spectroscopy measurements have documented large variation in the chemical composition of OM during petroleum generation. However, information regarding the heterogeneity of OM chemical composition at the nanoscale has been lacking. Here we demonstrate the first application of atomic force microscopy-based infrared spectroscopy (AFM-IR) to measure the chemical and mechanical heterogeneity of OM in shale at the nanoscale, orders of magnitude finer than achievable by traditional chemical imaging tools such as infrared microscopy. We present a combination of optical microscopy and AFM-IR imaging to characterize OM heterogeneity in an artificially matured series of New Albany Shales. The results document the evolution of individual organic macerals with maturation, providing a microscopic picture of the heterogeneous process of petroleum generation.

[1] Schlumberger-Doll Research Center, Cambridge, MA 02139, USA. [2] U.S. Geological Survey, Reston, VA 20192, USA. Correspondence and requests for materials should be addressed to A.E.P. (email: apomerantz@slb.com)

The recent boom in unconventional oil and gas production from shale has revolutionized the energy landscape in the United States[1] and stimulated both considerable research[2–5] as well as environmental and political concerns. Shale is a fine-grained sedimentary rock, composed of solid organic matter (OM) scattered in a mineral framework. This solid OM is the source of petroleum, as the OM breaks down into oil and gas under high temperatures during a process known as maturation. The solid OM remaining after maturation develops a network of nanoscale pores responsible for hydrocarbon storage and transport during petroleum production and migration over geologic time[6–8]. Additionally, OM partially controls shale's geomechanical properties as it is the softest component, affecting both fracturing efficiency and flow of petroleum to the wellbore[9]. Therefore, investigation of geochemical and geomechanical properties of OM in shale is important for building robust geological models to access unconventional resources and enhance hydrocarbon production efficiency.

Solid OM in shale is typically highly heterogeneous at the submicron length scale. Scanning electron microscopy (SEM) has shown that adjacent micron-sized OM grains sometimes differ greatly in porosity[10]. This heterogeneity can result from innate differences in the OM originally deposited in the shale and potentially from local variations in the extent of thermal alteration brought on by catalysis from nearby mineral grains. While SEM provides detailed, high-resolution information on pores in OM, it provides no information on the type, chemical composition, or mechanical properties of that OM. Organic petrography via incident light microscopy allows the identification of various types of organic components and has been broadly applied to shale petroleum systems[11], but again this technique provides no direct molecular or mechanical data. Numerous bulk measurements, such as programmed pyrolysis (Rock-Eval) and various spectroscopies[12] including infrared spectroscopy[13,14], have documented dramatic variations in the average chemical composition of OM across a wide maturation range, but these bulk analyses cannot assess heterogeneity at small spatial scales. Progress toward documenting geochemical heterogeneity in shale has been realized by application of Fourier transform infrared microscopy (micro-FTIR)[15–19]. Nevertheless, micro-FTIR cannot reliably resolve chemical features at the submicron length scale relevant to shale due to the diffraction limit[17,19,20]. Alternatively, Raman imaging provides structural information on the degree of organization of OM at the micron length scale but suffers from strong fluorescence background for immature shale samples[21–23]. Geomechanical heterogeneity in shale has been studied by nanoindentation at the micron length scale[24,25] and by atomic force microscopy (AFM) in peak force tapping mode at nanoscale[26,27], but these techniques provide no information regarding chemical composition.

AFM-based infrared spectroscopy (AFM-IR) is a rapidly emerging technique in materials and life sciences[28–34] that provides chemical and mechanical stiffness mapping at nanoscale unaffected by the diffraction limit, in additional to topographic imaging by AFM. During the measurement, an AFM probe with tip radius smaller than 25 nm is placed in contact with an area of interest simultaneously irradiated with a tunable IR laser[29]. When the laser wavelength is tuned to an infrared absorption band of the sample material, the sample thermally expands as the energy of absorbed photons is converted to heat. That thermal expansion acts as a force impulse on the AFM cantilever, driving it into oscillation. Oscillation amplitude is directly proportional to the local IR absorption coefficient at the IR wavelength at which the sample is irradiated. Because the AFM tip remains in contact with the sample, the contact resonance (CR) frequency of the coupled sample-cantilever system provides information about the

sample's mechanical stiffness, with higher frequency corresponding to stiffer materials and lower to softer materials (quantitatively translating the measured CR frequency to stiffness requires extensive modeling of the mechanical response of the cantilever to the photothermal expansion of the sample, so stiffness in AFM-IR measurements is typically described using the CR frequency)[28–30]. Thus, at a fixed laser wavelength, IR absorption and stiffness images can be collected simultaneously by recording the AFM cantilever oscillation amplitude and frequency. The spatial resolution is independent of the IR laser spot size, thus no longer limited by diffraction, but instead determined by the contact area between the probe and the sample as well as by the sample's thermomechanical properties[28–31,35]. The typical spatial resolution of the resulting chemical image is ~100 nm[28–31,35], orders of magnitude finer than the spatial resolution attainable by traditional micro-FTIR. Additionally, localized IR spectra can be obtained by measuring the AFM oscillation amplitude while tuning the IR laser wavelength. The resulting local IR absorption spectra are nearly identical to those measured with bulk transmission FTIR[28–31,35], providing information on chemical composition and structural characteristics.

In this study, we demonstrate the first application of AFM-IR in the earth sciences. This technique is used to provide direct, in situ, simultaneous geochemical and geomechanical characterization of individual dispersed OM particles at nanoscale. The AFM-IR images are correlated with optical microscopy images to analyze chemical and mechanical properties of macerals (optically discernible organic constituents) in a New Albany Shale sample. To understand the impacts of maturation on nanoscale OM composition, similar measurements are performed on a series of samples prepared to different stages of maturation by laboratory hydrous pyrolysis. This approach allows evaluation of compositional and mechanical variations among different macerals and unravels the chemical evolution of different macerals with increasing maturity. Beyond the bulk picture provided by conventional analysis, this approach of following compositional and mechanical evolution of individual macerals at the relevant length scale during maturation provides a microscopic picture of the heterogeneous process of petroleum generation.

## Results

**Shale rock assessment.** An OM-rich shale sample from Middle Devonian-Lower Mississippian New Albany Shale is chosen because it holds abundant, distinct, and readily identified macerals. The New Albany Shale sample is organic-rich, with total organic carbon (TOC) content of 14.2 wt.% (Table 1). The computed hydrogen index (HI) and oxygen index (OI) from Rock-Eval show the OM contains relatively high hydrogen/carbon ratio and low oxygen/carbon ratio. The sample is immature, with $T_{max} < 435\,°C$ and production index (PI) < 0.10, suggesting it is at the onset of oil generation[36]. This assessment agrees with solid bitumen reflectance ($BR_o$) of 0.25% as measured by optical microscopy. Solid bitumen reflectance is used here instead of conventional maturity indicators (e.g., vitrinite reflectance) owing to scarcity and difficulty in identification of vitrinite particles[20,37].

Hydrous pyrolysis of organic-rich shale at high temperature (320–360 °C) is used to simulate thermal maturation that takes place over millions of years in a sedimentary basin, providing a direct test of the impact of maturity, uncomplicated by other factors such as variation in mineralogy[38,39]. To understand how OM composition evolves with maturity, the immature New Albany Shale sample is pyrolyzed at temperatures ranging from 300 to 360 °C for 72 h, resulting in $BR_o$ values ranging from 0.41 to 1.17%. A linear relationship between pyrolysis

**Table 1 Organic geochemistry of the New Albany Shale sample and its hydrous pyrolysis derivative**

| Pyrolytic temperature (°C) | TOC | S1 | S2 | S3 | $T_{max}$ | HI | OI | PI | $BR_o$ (%) | Maturity stage |
|---|---|---|---|---|---|---|---|---|---|---|
| Unpyrolyzed | 14.2 | 5.0 | 71.7 | 0.8 | 429 | 504 | 6 | 0.06 | 0.25 | Prior to onset of oil generation (immature) |
| 300 | 13.1 | 4.9 | 63.4 | 0.7 | 434 | 483 | 6 | 0.07 | 0.41 | Prior to onset of oil generation (immature) |
| 320 | 11.6 | 4.0 | 27.6 | 0.3 | 436 | 237 | 2 | 0.13 | 0.65 | Early stage of oil generation (early mature) |
| 340 | 9.9 | 5.6 | 15.3 | 0.4 | 445 | 155 | 4 | 0.27 | 0.90 | Peak of oil generation (peak mature) |
| 350 | 8.7 | 2.9 | 6.9 | 0.4 | 454 | 79 | 4 | 0.29 | 1.09 | Late oil to early wet gas generation (late mature) |
| 360 | 8.9 | 2.2 | 5.2 | 0.5 | 472 | 58 | 5 | 0.30 | 1.17 | Late oil to early wet gas generation (late mature) |

Units: TOC in weight %; S1, S2 in mg hydrocarbon/g rock; S3 in mg $CO_2$/g rock; $T_{max}$ (temperature at which maximum yield of hydrocarbon occurs during pyrolysis) in °C. HI (hydrogen index) in mg hydrocarbons generated/g TOC; OI (oxygen index) in mg $CO_2$ generated /g TOC; PI (production index, S1/(S1 + S2)), unitless.
Data from Rock-Eval II pyrolysis, Leco TOC and solid bitumen reflectance ($BR_o$) analysis.
TOC, total organic carbon; S1, S2 and S3, are parameters determined in Rock-Eval pyrolysis, where S1 measures the amount of vaporized free hydrocarbons, S2 measures the amount of hydrocarbons generated through thermal cracking of nonvolatile organic matter, and S3 measures the amount of $CO_2$ produced during pyrolysis; HI, hydrogen index; OI, oxygen index; PI, production index

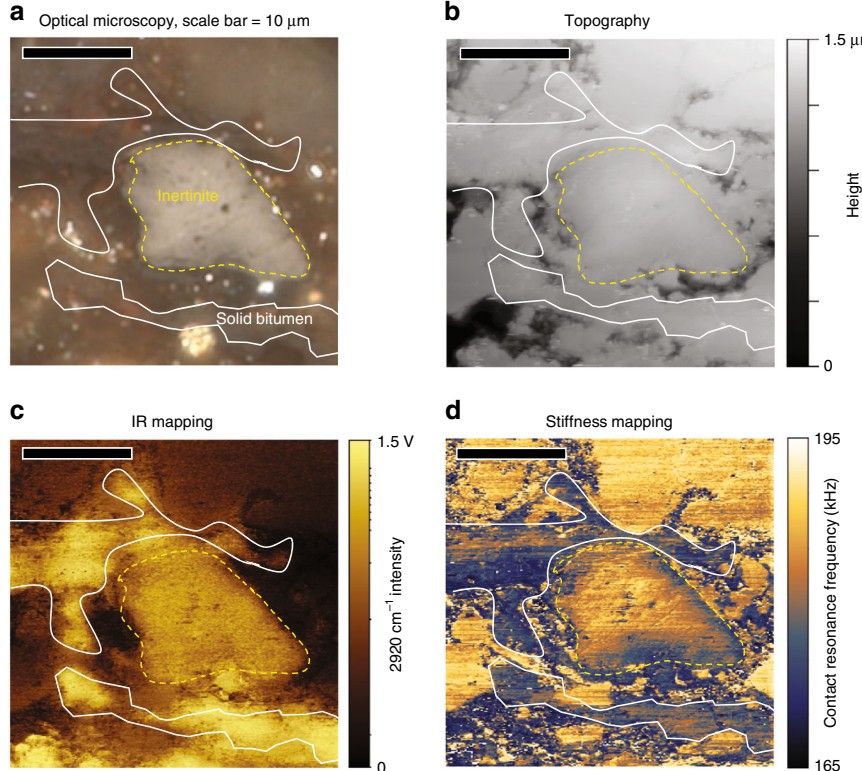

**Fig. 1** Correlative imaging of distinct inertinite and solid bitumen. **a** Photomicrograph (white incident light, oil immersion) of organic matter in polished, low maturity ($BR_o$, 0.25%) New Albany Shale sample, showing bright reflectance inertinite surrounded by solid bitumen in an scanned area of $30 \times 30 \, \mu m^2$. **b** Topographic image of same field, showing a smoothly polished surface, with no relief between organic and inorganic phases. **c** Same field showing IR absorption mapping of aliphatic C–H stretching (2920 cm$^{-1}$). **d** Mechanical stiffness mapping of same field at 2920 cm$^{-1}$

temperature and $BR_o$ (Supplementary Fig. 1) indicates $BR_o$ can be used to represent thermal maturity. The samples range from immature to late mature, including the onset of oil generation, early/peak oil generation, and late oil/early wet gas generation. TOC and Rock-Eval tests (Table 1) on the original New Albany Shale and its hydrous pyrolysis residues show systematic decrease in TOC, S2, and HI, as well as increase in $T_{max}$ and PI, with increasing hydrous pyrolysis temperature, indicating systematic increase in thermal maturity and conversion of OM to petroleum.

**Correlative AFM-IR and incident light microscopy imaging.** Macerals of interest in the New Albany Shale are imaged and identified by incident light microscopy and then registered and imaged by AFM-IR. In the unpyrolyzed (immature) sample,

abundant solid bitumen (gray solid hydrocarbon residue from amorphous organic matter conversion) and *Tasmanites* (a unicellular planktonic marine alga) are present, along with minor inertinite (remains of carbonized wood) fractions. For example, Fig. 1a shows the incident white light microscopy image, illustrating discrete solid bitumen and inertinite macerals from the unpyrolyzed immature New Albany Shale sample. The field of view in this image is $30 \times 30 \, \mu m^2$, comparable to a single pixel in a traditional micro-FTIR instrument. The AFM-IR measurements are performed in the same area, including images of high-resolution topography (Fig. 1b), IR absorption (Fig. 1c), and mechanical stiffness (Fig. 1d), all acquired with pixel size of $100 \times 100 \, nm^2$. The three AFM-IR images were simultaneously acquired by irradiating the sample with the IR fixed at 2920 cm$^{-1}$,

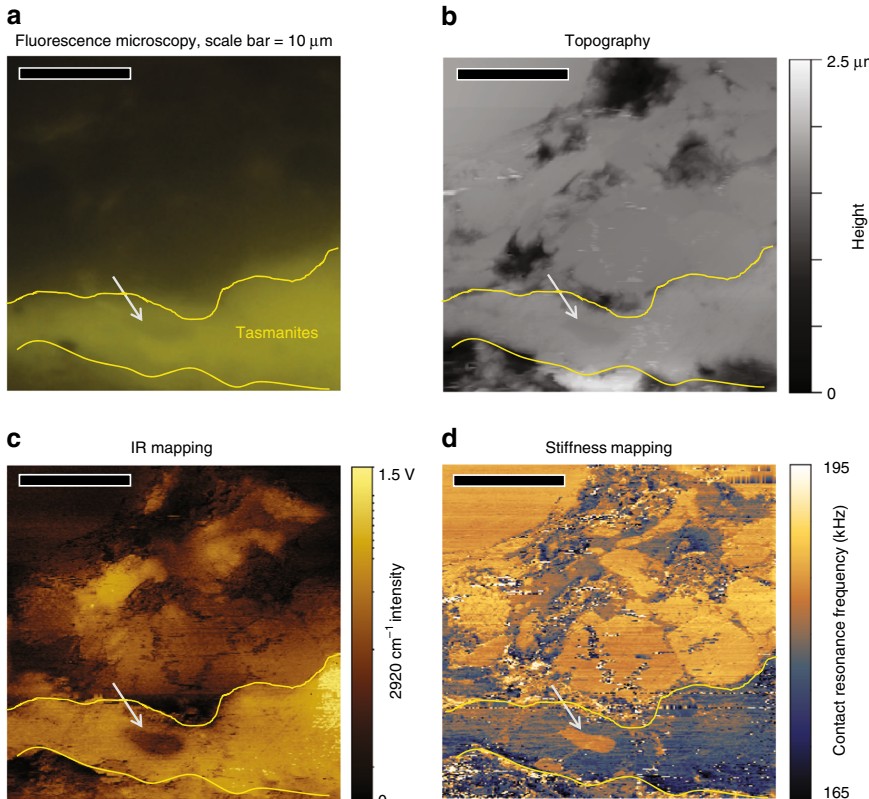

**Fig. 2** Correlative imaging of *Tasmanites*. Correlative imaging and atomic force microscopy infrared spectroscopy (AFM-IR) characterization, where the *Tasmanites* is outlined in yellow with one small mineral particle (indicated by the gray arrow) entrained inside: **a** fluorescence microscopy, showing brightly fluorescing *Tasmanites*, **b** topographic image, **c** IR absorption mapping at 2920 cm$^{-1}$, **d** mechanical stiffness mapping at 2920 cm$^{-1}$

characteristic of absorption by OM aliphatic C–H stretching. This feature[40] is conveniently distant from common absorption bands in inorganic phases[41,42], so imaging at this wavenumber enables interference-free mapping of OM. The topography image (Fig. 1b) shows relatively little contrast, indicating a smooth surface after mechanical polishing, such that OM cannot be identified. Instead, the discrete macerals are readily identified as having intense 2920 cm$^{-1}$ absorption (Fig. 1c) and low mechanical stiffness (Fig. 1d) relative to the surrounding mineral matter. The closely matching spatial distributions seen in the white light microscopy image and AFM-IR images confirm accurate registration.

In addition to the solid bitumen and inertinite macerals, the *Tasmanites* maceral in the immature New Albany Shale is identified and located by organic petrography (Fig. 2a). As before, the topography (Fig. 2b), IR absorption (Fig. 2c), and stiffness (Fig. 2d) images of the same area are collected with the AFM-IR by fixing the IR wavelength at 2920 cm$^{-1}$. Topography again indicates a well polished surface, with little distinction between OM and the inorganic matrix. IR mapping shows that *Tasmanites* strongly absorbs at 2920 cm$^{-1}$, whereas the small inorganic particle (~1 μm across) entrained inside (indicated by the gray arrow) does not absorb. The stiffness image reveals significant contrast between *Tasmanites*, the embedded particle, and surrounding mineral matrix. The embedded particle has similar stiffness to the mineral matrix and does not absorb at 2920 cm$^{-1}$, confirming its assignment as mineral matter. These observations demonstrate the capability of documenting spatial variability in chemistry and mechanical properties of shale at nanoscale with AFM-IR.

**Geomechanical and geochemical variations among macerals.** Geomechanical heterogeneity can be estimated by analyzing the

distribution of contact resonance (CR) frequencies measured on each maceral, as CR frequency is positively correlated with sample stiffness. Inertinite, solid bitumen, and *Tasmanites* are highlighted in stiffness images Fig. 3a–c, respectively. Prior to the AFM-IR imaging, the AFM probes are carefully selected to have nearly the same contact resonance frequency when measured on a polystyrene thin film sample. Figure 3d displays the distribution of CR frequencies from pixels assigned as inertinite (highlighted in Fig. 3a), where each pixel (100 × 100 nm$^2$) contributes one data point. Inertinite accounts for an area of 130.74 μm$^2$, with 13,074 measured points yielding an average CR frequency of 182.1 kHz ± 1.7 kHz (Table 2). Figure 3e, f displays the resulting CR frequency distributions of solid bitumen (178.8 kHz ± 3.6 kHz) and *Tasmanites* (177.2 kHz ± 4.3 kHz), indicating the average mechanical stiffnesses of *Tasmanites* is slightly less than solid bitumen and much lower than inertinite.

Geochemical heterogeneity can be probed by measuring the IR spectra collected at different locations corresponding to different macerals. Localized IR spectra are obtained for inertinite, solid bitumen, and *Tasmanites*, as indicated by the colored dots in Fig. 3a-c. Because the absolute intensities of localized IR spectra are affected by absorbing material thickness, all spectra are normalized for comparison. All localized spectra illustrate the characteristic bands of IR absorption in OM: aliphatic absorption at 2920 cm$^{-1}$, 2850 cm$^{-1}$, 1450 cm$^{-1}$, and 1375 cm$^{-1}$; aromatic absorption at 3100–3000 cm$^{-1}$ and 1600 cm$^{-1}$; and carboxyl absorption at ~1710 cm$^{-1}$ (details of band assignments for IR spectra of OM are described in the "Methods" section). Different locations within a maceral present nearly identical IR spectra, as observed for the inertinite (Fig. 3g), solid bitumen (Fig. 3h), and *Tasmanites* (Fig. 3i) macerals. These results indicate compositional homogeneity within a maceral type. However, large

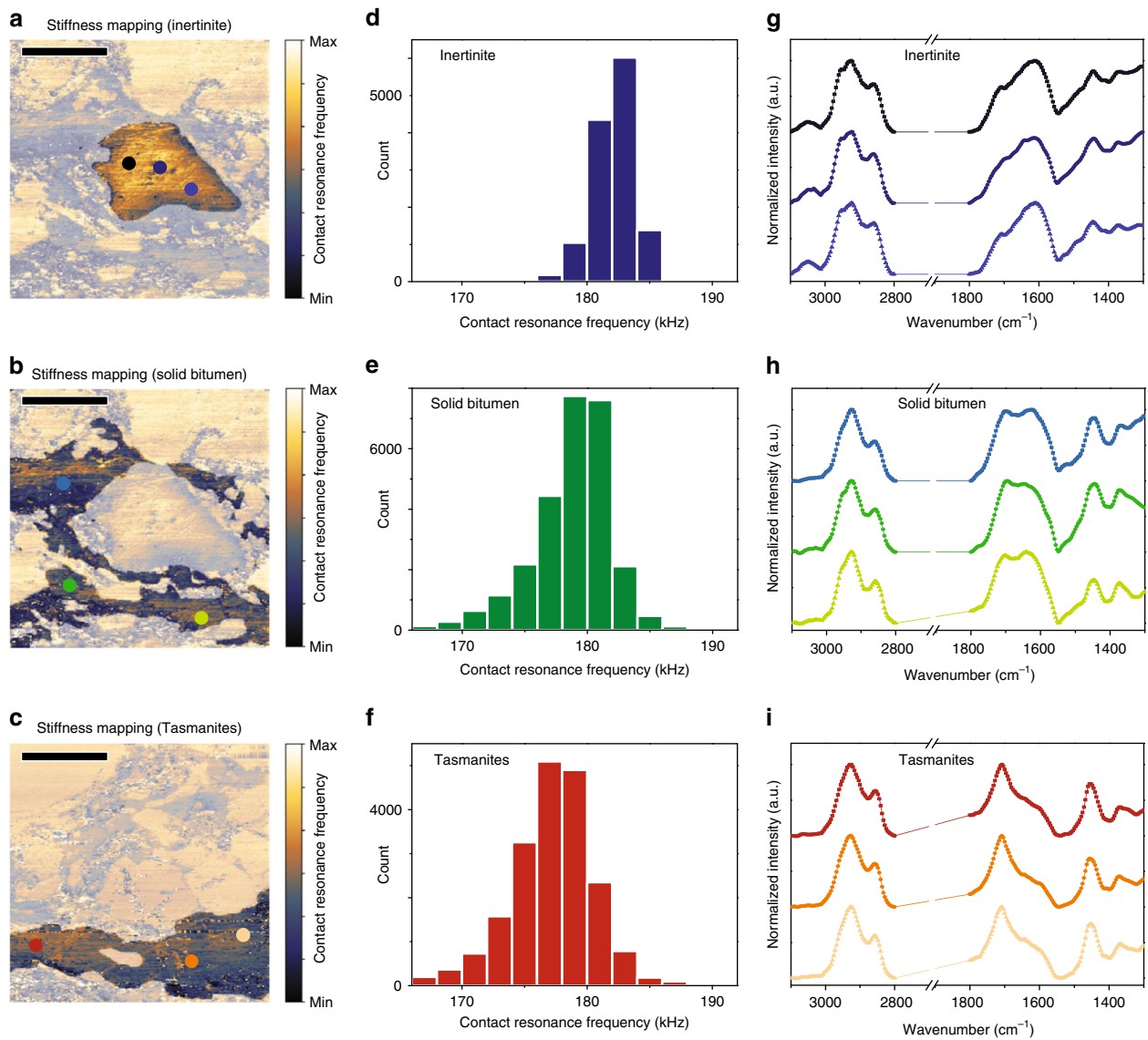

**Fig. 3** Nanoscale geochemical and geomechanical characterization. Inertinite (**a**), solid bitumen (**b**), and *Tasmanites* (**c**) in stiffness images are identified by comparison with photomicrograph of the same field (Figs. 1a and 2a). Areas outside of regions of interest are masked with opaque color to screen out irrelevant contact resonance (CR) frequency measurements. CR frequency histogram for 130.74 μm$^2$ area of inertinite (**d**), 272.97 μm$^2$ area of solid bitumen (**e**), and 201.67 μm$^2$ area of *Tasmanites* (**f**). Localized IR spectra from different locations (colored points in corresponding stiffness images) in inertinite (**g**), solid bitumen (**h**), and *Tasmanites* (**i**)

**Table 2 IR ratios and mechanical stiffnesses computed from the average localized AFM-IR spectra from inertinite, solid bitumen, and *Tasmanites* in immature New Albany Shale**

| Maceral-specific chemical composition and mechanical properties | Aromaticity | CH$_3$/CH$_2$ | C-factor | A-factor | Average contact resonance frequency (kHz) |
|---|---|---|---|---|---|
| Inertinite | 0.1 | 1.97 | 0.29 | 0.37 | 182.1 ± 1.7 |
| Solid bitumen | 0 | 1.24 | 0.42 | 0.41 | 178.8 ± 3.6 |
| *Tasmanites* | 0 | 0.63 | 0.70 | 0.57 | 177.2 ± 4.3 |

variations exist between maceral types in immature shale (see also Supplementary Fig. 2). Inertinite displays a strong aromatic character, exhibiting a distinct aromatic absorbance band at 3000–3100 cm$^{-1}$, strong aromatic absorbance at 1600 cm$^{-1}$, and relatively low oxygenated group stretching at ~1710 cm$^{-1}$. *Tasmanites* shows intense C=O absorption at ~1710 cm$^{-1}$ and low aromatic absorption at 1600 cm$^{-1}$. Solid bitumen shows no

aromatic C–H absorbance at 3000–3100 cm$^{-1}$, modest aromatic C=C ring stretching at 1600 cm$^{-1}$ and modest C=O stretching at ~1710 cm$^{-1}$.

IR ratios from the average localized IR spectra from inertinite, solid bitumen, and *Tasmanites* in immature New Albany Shale are computed to evaluate quantitatively the chemical differences between OM types (Table 2). Four IR ratios are computed, similar

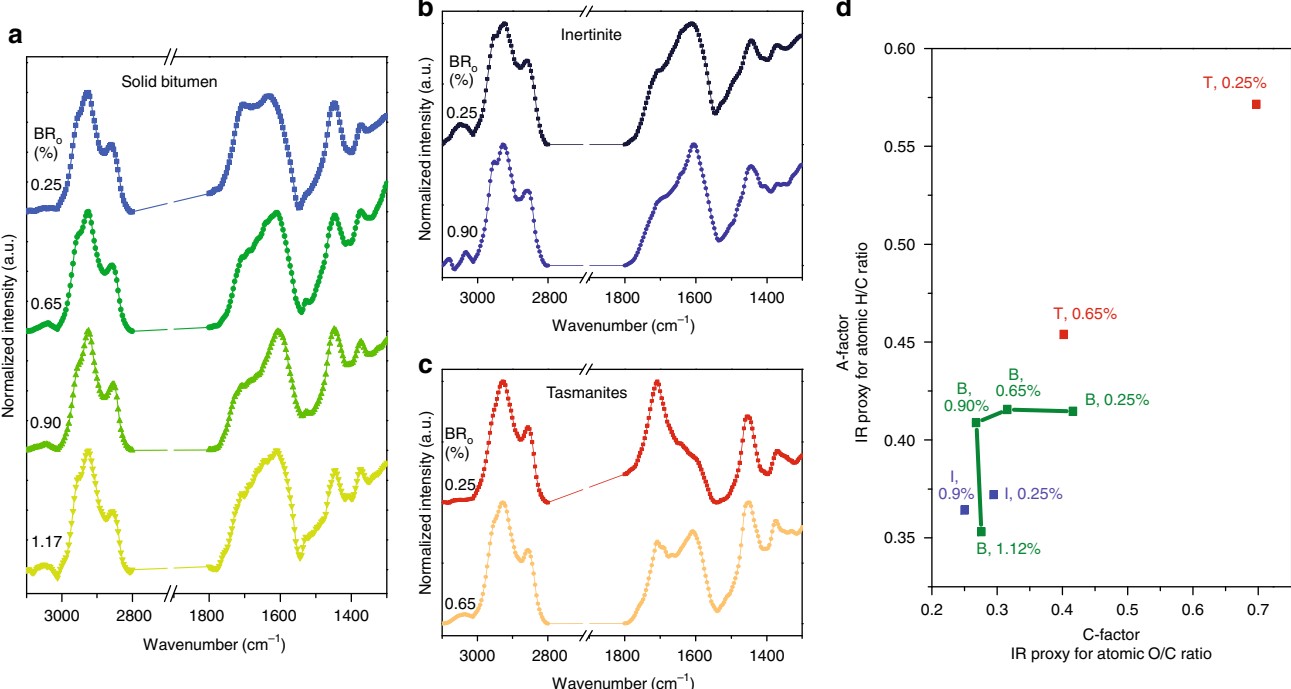

**Fig. 4** Compositional trends in the maturation of individual macerals. **a** Average localized infrared (IR) spectra from three or more different solid bitumen particles in untreated and artificially matured New Albany Shale samples, with solid bitumen reflectance (BR$_o$) ranging from 0.25% to 1.17%. **b** Average localized IR spectra from different locations in 1–2 inertinite particles in samples with BR$_o$ of 0.25 and 0.90%. **c** Average localized IR spectra from different locations in 1–2 *Tasmanites* particles in samples with BR$_o$ of 0.25 and 0.65%. **d** Pseudo van Krevelen diagram, constructed by plotting the IR proxy for H/C and O/C atomic ratios of *Tasmanites*, solid bitumen, and inertinite at different maturities. Solid bitumen reflectance values are indicated adjacent to calculated IR ratios (T, *Tasmanites*; B, solid bitumen; and I, inertinite)

to those defined in the bulk IR literature[14,17,43]: aromaticity (aromatic C–H versus aliphatic C–H stretching modes), CH$_3$/CH$_2$ (methyl-to-methylene ratio, an indicator of aliphatic chain length and/or the degree of branching of aliphatic moieties), C-factor (proportional to the oxygenated character), and A-factor (proportional to the aliphatic character). Results (Table 2) show that the various OM types in the New Albany Shale are distinct in terms of their chemical composition. *Tasmanites* is relatively rich in oxygenated functional groups and aliphatic carbon, with relatively long chains and little aromatic carbon. Inertinite lies on the other extreme, dominated by aromatic carbon with relatively short chains and relatively little oxygenated functional groups or aliphatic carbon. Solid bitumen is intermediate in all regards, with moderate amounts of aliphatic carbon, aromatic carbon, oxygenated functional groups, and chain length. The mechanical stiffness (Table 2) is correlated with the chemical composition of OM, showing increasing stiffness with higher aromaticity, similar to other findings[4,27].

**Evolution of chemical composition of individual macerals.** Maturation is the process under which petroleum is generated by thermal cracking of OM at elevated temperature. The combination of organic petrography with AFM-IR allows the maturation process to be followed on a maceral-by-maceral basis, supplementing the traditional analysis of bulk chemical changes in OM[14,43]. Here the New Albany Shale is artificially matured by hydrous pyrolysis to various stages in the petroleum generation process, and the evolution in the abundance of individual macerals is measured by petrography while the evolution in the composition of individual macerals is measured by AFM-IR.

Initially, immature New Albany Shale holds abundant solid bitumen and *Tasmanites*, with a small fraction of inertinite. During maturation, the abundance of inertinite remains

qualitatively unchanged. However, the abundance of *Tasmanites* and solid bitumen change dramatically during maturation. Similar to petrographic observations in other shale plays[11], *Tasmanites* is observed to decrease in abundance throughout the process, finally disappearing at HP temperatures ≥340 °C, while solid bitumen becomes the dominant maceral at high temperatures.

AFM-IR is used to measure to composition of individual macerals, such as solid bitumen, at different maturities (see example of correlative imaging of micron-scale solid bitumen particles in pyrolyzed shale in Supplementary Fig. 3). Localized IR spectra averaged over three or more different solid bitumen particles in each of the original New Albany Shale sample and its hydrous pyrolysis derivatives are plotted in Fig. 4a (those individual spectra are plotted in Supplementary Figs. 4–7). The A-factor (aliphatic versus aromatic) and C-factor (oxygenated versus aromatic functional groups) are calculated from the average spectra and used as IR proxies for atomic H/C and O/C ratios, respectively[43]. Figure 4d plots these parameters similar to a standard van Krevelen diagram, with green points labeled by maturity as measured by solid bitumen reflectance (BR$_o$). The plot illustrates the evolution path of solid bitumen during thermal alteration, showing early loss of oxygenated functional groups followed by loss of aliphatic content at higher pyrolysis tempreatures, consistent with trends observed previously in bulk OM[43]. A similar analysis was perfomed to investigate the chemical evolution of inertinite (Fig. 4b) and *Tasmanites* (Fig. 4c) with maturity. Direct comparison of the compositional evolution of all macerals is shown in the pseudo van Krevelen diagram (Fig. 4d). Significant differences between the composition of these macerals are seen at low maturity, but those compositional differences become less significant at higher maturity when the composition of each maceral type tends toward graphite-like

(high aromatic content and low aliphatic and oxygenated content). Macerals that are rich in aliphatic/oxygenated content at low maturity (such as *Tasmanites*) undergo relatively large compositional changes with maturity, while macerals that are already dominated by aromatic carbon at low maturity (such as inertinite) undergo relatively small compositional changes with maturity. For example, a relatively large difference is found between the composition of *Tasmanites* at 0.25% $BR_o$ and at 0.65% $BR_o$ (the highest maturity at which *Tasmanites* is detected), consistent with the high petroleum generation potential of *Tasmanites*. Inertinite, on the other hand, is mostly unaffected by maturation, with little change in its abundance or composition. Moreover, the range of compositions spanned by a single maceral at different thermal maturities is comparable to the range of compositions spanned by different macerals at a single (low) thermal maturity. This observation suggests that the impact of maturation in OM composition is driven by both an evolution of the abundance of different macerals and a variation in the composition of individual macerals.

In conclusion, here we demonstrate AFM-IR as a powerful new tool to examine organic chemical and mechanical heterogeneity in shale at nanoscale. AFM-IR images can be registered with traditional optical microscopy images, allowing geochemical and geomechanical characterization of individual macerals assigned by organic petrography. The measurement provides chemical and mechanical mapping at a spatial resolution orders of magnitude finer than provided by traditional diffraction-limited IR microscopy, enabling interrogation of chemical and mechanical heterogeneity at the submicron length scales relevant for shale. Using this approach, we examined the evolution of OM composition during petroleum generation at the maceral level. Immature shale contains several different macerals and the average composition of each maceral type differs widely, although the compositional variation between different locations belonging to the same maceral type is limited. The composition of the macerals controls their mechanical properties, with macerals enriched in aromatic carbon (and lean in aliphatic carbon) having relatively high mechanical stiffnesses. During maturation, both the abundance and the composition of the individual macerals evolve. Organic petrology revealed that originally present macerals, such as *Tasmanites*, are depleted during petroleum generation, and the OM in shale is dominated by the solid bitumen maceral at maturities above the peak oil level. AFM-IR revealed that the composition of each maceral evolves as well, as each maceral becomes enriched in aromatic carbon and depleted in oxygenated carbon during maturation, similar to the trend commonly observed for bulk analysis of OM. The results suggest the maturation process involves an evolution in both the abundance of individual macerals and in the composition of those macerals.

## Methods

**Hydrous pyrolysis**. Hydrous pyrolysis followed the method of Lewan[44,45]. The crushed rock samples (2–4 g, 1–3 mm maximum size) were loaded into gas chromatograph ovens and isothermally heated for 72 h, plus ~5–10 min warm-up. Rock residues were rinsed in acetone at experiment conclusion to remove generated bitumens from surfaces. Residues were vacuum-dried overnight prior to preparation for petrographic and AFM-IR analysis.

**Organic petrography analysis via optical microscopy**. Organic petrographic techniques included maceral identification in epi-fluorescence and reflected white light under oil immersion, and reflectance measurements of solid bitumen for maturity estimation. The New Albany Shale rock powder and its pyrolysis residues (~1 mm size) were mounted in heat-setting plastic 1-inch (25.4 mm) round pellets and observed using a reflected light microscope (Zeiss) with oil immersion objectives (×50, 1.0 NA; ×100, 1.30 NA). Sample preparation followed ASTM D2797[46] wherein thermoset plastic pellet sample mounts were mechanically ground and polished by paper (120–1200 grit) using water as lubricant followed by final polish on matte clothes with 1.0 µm alumina and 0.05 colloidal silica abrasives

to achieve smooth surfaces for microscopic and AFM-IR analysis. Maceral identification and solid bitumen reflectance measurements followed ASTM D7708[47] as previously described[11]. Reflectance analysis employed a Leica DM4000 microscope with LED illumination and monochrome camera detection with the computer program DISKUS-FOSSIL by Hilgers Technisches Buero and a Klein and Becker YAG calibration standard (0.908% $R_o$). Reflectance is a measurement of the percentage of light reflected from macerals under oil immersion. For the New Albany Shale sample and its pyrolysis residues, reflectance of 30–50 random individual solid bitumen particles was measured and the mean value reported as $BR_o$.

**Bulk characterization of shale**. Total organic carbon (TOC) analysis and Rock-Eval II (a programmed pyrolysis method) are routine source rock geochemistry characterization methods, allowing evaluation of the amount, type and thermal maturity of organic matter present. TOC and Rock-Eval were carried out in a commercial laboratory following methods previously described in Barker[48] and Peters et al.[36] Leco TOC measurements were performed by combustion of acidified pulverized rock to quantify organic richness. Rock-Eval II pyrolysis measures hydrocarbon generating potential and thermal maturity. Rock-Eval heats samples in programmed stages ranging from 100 to 850 °C, yielding four parameters: S1 measures the amount of vaporized free hydrocarbons, S2 measures the amount of hydrocarbons generated through thermal cracking of nonvolatile organic matter, S3 measures the amount of $CO_2$ produced during pyrolysis, and $T_{max}$ (pyrolysis temperature during maximum generation of hydrocarbons). S1, S2, and S3 are used to compute hydrogen, oxygen, and production indices [HI: $100 \times S2/TOC$; OI: $100 \times S3/TOC$; PI: $S1/(S1 + S2)$]. HI and OI determine the organic matter type, whereas $T_{max}$ and PI represent the thermal maturity of organic matter.

**AFM-IR measurements**. The localized nanoscale mid-IR spectra and images were carried out using the commercial NanoIR2 AFM-IR instrument (Anasys Instruments). The AFM-IR technique is accomplished by coupling a pulsed tunable IR source with an AFM. The IR source covers a broad range of the mid-IR region, produced by a tunable optical parametric oscillator laser system (EKSPLA) with ~10 ns pulse length at a repetition rate of 1 kHz.

The topography, IR absorption, and contact resonance frequency image (stiffness map) are collected simultaneously at a fixed wavenumber of interest. All images are acquired in contact mode using probes (Model: PR-EX-NIR2, Anasys Instruments) with a resonance frequency of $13 \pm 4$ kHz and a spring constant of $0.07–0.4$ N m$^{-1}$. The characterization measures the photothermal-induced resonance of the AFM cantilever[28–31]. Here, the second mode of cantilever oscillation is selected by applying a bandpass filter centered at 180 kHz with 50 kHz window. The images are acquired at a scan rate of 0.2 Hz and averaged over 8 pulses, with a 100 nm × 100 nm spatial resolution. The image acquisition time is ~25 min for a 30 × 30 µm area or 11 min for a 20 × 20 µm area. The spatial resolution of IR absorption imaging is affected by sample thermomechanical properties such as thermal expansion coefficient and thermal conductivity[28–31]. The typical spatial resolution of IR images is around 100 nm. Stiffness mapping offers enhanced resolution limited by the tip radius of the cantilever, typically in the range of 10 s of nanometers.

**Spectral analysis and IR structural indices calculation**. Localized IR spectra were collected and analyzed using Analysis Studio software (version 3.11.5883, Anasys Instruments) and MATLAB program. IR focus is optimized at 1450 cm$^{-1}$, 1600 cm$^{-1}$, and 2920 cm$^{-1}$ prior to the data collection and all IR spectra are normalized to the incident power. IR spectra are acquired from 1300 to 1800 cm$^{-1}$ and 2800 to 3100 cm$^{-1}$ with an interval of 4 cm$^{-1}$ and averaging each data point over 256 pulses. The spectral acquisition time is around 5 min. The localized IR spectra is first smoothed using a three-point average in Analysis Studio, followed by linear baseline removal, normalized, and plotted in MATLAB.

Curve fitting of the absorbance bands in IR spectra between 1500–1800 cm$^{-1}$ and 2800–3100 cm$^{-1}$ were performed to determine spectral parameters in MATLAB. A combination of Lorentzian–Gaussian curve functions provides a good statistical reconstruction of the measured IR spectra of complex organic matter[43]. The parameters are fitted to the experimental envelope by a least squares iterative procedure. All peaks centers, peak amplitudes, peak half-widths, relative fraction of Lorentzian character for the fit are allowed to vary from initial guesses. Generally, the solved peak centers are typically within a few centimeters from the initialization. The IR structural parameters, namely, Aromaticity, CH$_3$/CH$_2$, C-Factor, A-Factor, are computed from the peak area ratios as listed in Supplementary Table 1.

The spectral region at 2800–3000 cm$^{-1}$ can be resolved into five discrete bands: CH stretching (2897 cm$^{-1}$), symmetric CH$_2$ stretching (2857 cm$^{-1}$), CH$_2$ asymmetric stretching (2925 cm$^{-1}$), CH$_3$ symmetric stretching (2872 cm$^{-1}$), and CH$_3$ asymmetric stretching (2962 cm$^{-1}$). The broad absorbance band between 3000 and 3100 cm$^{-1}$ region is assigned to aromatic C–H stretching and used to compute Aromaticity. CH$_3$/CH$_2$ ratio are computed from the ratio of integrated peak areas arising from 2957 and 2925 cm$^{-1}$, respectively, i.e., $A_{2957}/A_{2925}$.

The 1500–1800 cm$^{-1}$ region contains overlapping absorbance bands associated with oxygenated and aromatic carbon groups. The broad absorbance 1600–1630 cm$^{-1}$ corresponds to aromatic carbon–carbon stretching, and it can be resolved as

two distinct peaks in the spectra of organic materials centered at 1610 and 1630 cm$^{-1}$. A$_{1600-1630}$ represents the integrated peak intensity of the aromatic C=C vibrations centered at 1610 and 1630 cm$^{-1}$. The 1450 and 1375 cm$^{-1}$ absorption signals are due to aliphatic C–H bending. C-factor represents the intensity ratio of the oxygenated versus aromatic functional groups. A-factor is the intensity ratio of aliphatic/ (aliphatic + aromatic) bands. A-factor is calculated slightly different from Craddock's method[43]. The integrated peak area under 1450 cm$^{-1}$ is used instead of the area under 2857 and 2925 as the aliphatic absorption bands. This is because the pulsed optical parametric oscillator (OPO) laser used in the nano-IR instrument has two distinct stages. The laser spot size and focus is different in the different stages, thus it is difficult to compute quantitative IR ratios using intensity values from two different stages.

**Data availability**. The data that support the findings of this study are available from the authors on reasonable request (see author contributions for specific data sets).

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

## Acknowledgements

Reviews by Aaron Jubb (USGS), Robert L. Kleinberg (Schlumberger), and John Ratulowski (Schlumberger) improved this manuscript. Peter Warwick (USGS)

contributed the New Albany shale sample. Paul R. Craddock (Schlumberger) provided suggestions toward spectral analysis. This research was funded in part by the USGS Energy Resources Program. Any use of trade, firm, or product names is for descriptive purposes only and does not imply endorsement by the U.S. Government.

## Author contributions

A.P. developed the concept of AFM-IR characterization of shale samples, J.Y. conceived and designed the experiments, and P.H. conceived the approach of sample evaluation across thermal gradients. J.H. and P.H. polished and prepared the New Albany shale sample, measured the bulk chemistry with TOC and Rock-Eval, and identified and imaged maceral types using optical microscopy. J.Y. performed the AFM-IR imaging and localized IR experiments and together with A.P. developed spectral analysis. All authors discussed the data. J.Y. wrote the manuscript with support from P.H. and A.P.

## Additional information

**Competing interests:** The authors declare no competing financial interests.

