## [Peer Review File · Nature Communications]

Reviewers' comments:

Reviewer #1 (Remarks to the Author):

Review of "Nanoscale geochemical and geomechanical characterization of dispersed organic matter in shale"

In the paper the authors use a new atomic force microscopy technique (AFM-IR) to chemically and mechanically map the different organic components in an organic rich shale at the nanometer scale. The authors demonstrate that different organic components (bitumen, intertinite, and Tasmanites) exhibit distinct chemical signatures which are attributed to the proportions of different functional groups. Moreover, by carrying out pyrolysis experiments the authors show that these different components undergo contrasting changes during maturation. Their results show the potential that such measurements have in the study of organic matter in shales, and how they can be used to track changes to individual macerals. I think the paper is timely, fascinating, of high quality, and I recommend its publication in Nature Communications. Below I list some minor comments that I think could help improve the impact of the study.

1. The main strength and novelty of the paper is the chemical characterization of the organic material. What's the correlation (on a pixel to pixel basis) between the chemistry and the mechanical properties? Could such an analysis be used to say something about the high degree of heterogeneity reported for the mechanical properties for individual organic macerals (e.g., Emmanuel et al., 2016). This might be worth discussing.
2. As presented in the paper, the mechanical properties are qualitative. Is it possible to calibrate the reported frequencies so that they could be reported as elastic moduli?
3. Can the authors discuss in greater detail (at least qualitatively) how the mechanical properties evolve during maturation?
4. Why did the authors choose mechanical polishing rather than argon milling? The IR technique will be sensitive to smearing of organic matter and this could potentially impact the results.

A number of specific comments are listed below:

5. Line 2: In the title the word “dispersed” seems unnecessary.
6. Line 16: The last sentence of the abstract is vague. Be more specific.
7. Lines 20-21: In the phrase “composed of dispersed organic matter (OM) scattered in a mineral framework”, the either “dispersed” or “scattered” is redundant.
8. Line 40: These methods can assess heterogeneity, just not at small spatial scales.
9. Lines 57-59: What is meant by the stiffness? Is this the elastic modulus? Is there a specific relationship between the frequency and modulus?
10. Lines 75-76: The method as presented by the authors is - for mechanical properties at least – qualitative and not quantitative.

Reviewer #2 (Remarks to the Author):

The paper presents the results of a study on the geomechanics and geochemistry of organic matter as a function of maturity at nano-scale. The topic is important and the usage of the methodology at nano-scale is novel. However I found several weaknesses in the study which requires major revision to the manuscript.

Major comments:

-The authors claim characterization of mechanical properties of organic matter at small scales. However, the stiffness maps they have provided just represent qualitative mechanical mapping and not quantitative characterization. The higher stiffness of macerals with higher concentration of

aromatic carbon has been known, but the stiffness value for different types of macerals is not known which requires quantitative measurement.

- The authors examined the evolution of OM composition with maturation. However, they examined only one type of shale: New Albany shale. Chemical composition of OM is dependent on depositional environment and the results obtained from studying the evolution of OM within one type of shale can not be generalized to others. Thus performing the same type of analysis on different samples will result in better understanding of the impact of maturation on nanoscale OM composition.

- The authors did not explain about the calibration of spring constant which is an essential step in nanomechanical mapping.

- Many findings listed in the conclusion have been obtained by traditional measurements (such as enrichment of macerals with aromatic carbon with maturation). So one of the contributions of this study would be the distribution of different macerals within OM and its evolution with maturation.

Reviewer #3 (Remarks to the Author):

The author has presented a pioneering study of shale by the AFMIR technique. They have used optical microscopy and nanoscale infrared imaging to characterize the solid organic matter into shale rock for different maturations. They have clearly demonstrated the ability of the AFMIR technique to follow the evolution of maturation providing a new sight of the microscopic process of petroleum generation. This paper opens a new field of application for infrared nanoscale technique.

In Conclusion, I recommend to accept this paper for publication with minor corrections.

1)Page3-line64. The resolution of AFMIR technique is not 100 nm. The resolution is just limited by the radius of the tip and can achieved a few nanometers (ref30).

2)Page5-line117. The authors explain that the topography shows little contrast. Do the authors could make tapping mode image to check if the same region shows the same topography? This would just confirm that the stiffness of the surface do not disturb the topography.

3)Page6-line119. Authors speak about low mechanical stiffness. Could they explain this part more precisely. The frequency shift of the resonance is consistent only if it is associated to absorption.

4) page6-line134. What is the idea behind quantified. Does that mean that the authors can measure the absolute stiffness of each part of the shale?

5)Page7-line151. Does the spectra obtained by AFMIR could be compared with FTIR? Is there a way to confirm that the AFMIR spectra are consistent?

6)Page12-line284. Does the authors has checked the heterogeneity with smaller scan like 3x3 um?

7)Page13-line287. As explained previously the resolution is not 100 nm.

8) Page23-line474. There is no value for absorption and frequency maps, only min and max. Even if IR mapping is in arbitrary unit, it is really useful to know if the min value correspond to zero absorption. I suggest that the author add values to both maps.

9) Page24-line485. Same as 8).

10)Page27-line518. Does the spectra are taken at the same position after maturation?

Response to Reviewer 1

Overall Comments:

- *“In the paper the authors use a new atomic force microscopy technique (AFM-IR) to chemically and mechanically map the different organic components in an organic rich shale at the nanometer scale. The authors demonstrate that different organic components (bitumen, intertinite, and Tasmanites) exhibit distinct chemical signatures which are attributed to the proportions of different functional groups. Moreover, by carrying out pyrolysis experiments the authors show that these different components undergo contrasting changes during maturation. Their results show the potential that such measurements have in the study of organic matter in shales, and how they can be used to track changes to individual macerals. I think the paper is timely, fascinating, of high quality, and I recommend its publication in Nature Communications. Below I list some minor comments that I think could help improve the impact of the study”*

General Questions

1. *“The main strength and novelty of the paper is the chemical characterization of the organic material.”*
 - a. *What’s the correlation (on a pixel to pixel basis) between the chemistry and the mechanical properties?”*
 - b. *“Could such an analysis be used to say something about the high degree of heterogeneity reported for the mechanical properties for individual organic macerals (e.g., Emmanuel et al., 2016). This might be worth discussing.”*

Response:

- a. We find that both the chemistry and the mechanical properties are relatively constant on a pixel to pixel basis if all the pixels belong to the same maceral, while large variations occur between macerals (Figure 3). As a result, correlations between mechanical properties and chemistry on a pixel to pixel basis within a maceral may be influenced by measurement noise, while we can be more confident in such correlations on a maceral to

maceral basis. On the maceral basis, we find that higher aromatic content is correlated with mechanically stiffer structures (Table 2).

- b. Yes, such an analysis could be quite interesting, although beyond the scope of this work. The method employed by Emmanuel provides quantitative mechanical information and no chemical information, while the method employed here provides qualitative mechanical information and quantitative chemical information. By combining the two quantitative measurements, it may be possible to get past the noise issue discussed above and investigate quantitative mechanical and chemical heterogeneity on a pixel by pixel basis.

2. *“As presented in the paper, the mechanical properties are qualitative. Is it possible to calibrate the reported frequencies so that they could be reported as elastic moduli?”*

Response: Unfortunately it is not yet possible to translate the reported frequencies to elastic moduli. Several groups are working on the subject at the moment (to our knowledge), but converting the contact resonance frequencies to elastic moduli is theoretically complex and challenging to execute; we are not aware of any paper reporting quantitative elastic moduli based on the measured contact resonance frequencies as of today.

3. *“Can the authors discuss in greater detail (at least qualitatively) how the mechanical properties evolve during maturation?”*

Response: The evolution of mechanical properties with maturity has been discussed previously, for example by Emmanuel et al., 2016[1], using an instrument that provides quantitative mechanical information and no chemical information. Performing a similar measurement using the technique employed here, which simultaneously provides quantitative chemical information, is experimentally challenging for several reasons. Thus, this work focuses on how the chemical properties at nanoscale evolve with maturity, which previously has not been possible to explore.

4. *“Why did the authors choose mechanical polishing rather than argon milling? The IR technique will be sensitive to smearing of organic matter and this could potentially impact the results.”*

Response: Mechanical polishing is part of standard procedures for preparing pellets for organic petrographic studies (the sample preparation used here followed ASTM D2797 as described in detail in the methods section). Argon ion milling is a potential alternative that has advantages for some imaging experiments such as electron microscopy. However, recent reports suggest argon milling potentially can alter the organic matter chemistry [Sanei, et al. 2016, Alteration of organic matter by ion milling, International Journal of Coal Geology]. Because AFM-IR is sensitive to organic matter chemistry, and because data of sufficient quality could be obtained using mechanically polished samples, we found it prudent to avoid argon ion milling.

Part Two: Specific comments

5. *“Line 2: In the title the word “dispersed” seems unnecessary”*

Changes we made to the main text regarding this question:

- We deleted “dispersed” from title.

6. *“Line 16: The last sentence of the abstract is vague. Be more specific”*

Response: The tight word limit in the abstract makes it difficult to elaborate on this point. Hopefully the abstract is intriguing enough to direct the reader to the main text!

7. *“Lines 20-21: In the phrase “composed of dispersed organic matter (OM) scattered in a mineral framework”, the either “dispersed” or “scattered” is redundant.”*

Changes we made to the main text regarding this question:

- We deleted “dispersed” from the sentence.

8. *“Line 40: These methods can assess heterogeneity, just not at small spatial scales.”*

Changes we made to the main text regarding this question:

- We added these methods cannot assess heterogeneity “at small spatial scales”.

9. *“Lines 57-59: What is meant by the stiffness? Is this the elastic modulus? Is there a specific relationship between the frequency and modulus?”*

Response: Stiffness is a qualitative description of mechanical properties that is related to the elastic modulus and also to other properties such as physical dimensions. The contact resonance

frequency has an S-shaped dependence on the sample's elastic modulus. We refer the reviewer to literature describing general application of AFM-IR for details [2].

10. *“Lines 75-76: The method as presented by the authors is - for mechanical properties at least – qualitative and not quantitative.”*

Changes we made to the main text regarding this question:

- We deleted “quantitative”.

Response to Reviewer 2

Overall Comments:

- *“The paper presents the results of a study on the geomechanics and geochemistry of organic matter as a function of maturity at nano-scale. The topic is important and the usage of the methodology at nano-scale is novel. However I found several weaknesses in the study which requires major revision to the manuscript.”*

1. *“The authors claim characterization of mechanical properties of organic matter at small scales. However, the stiffness maps they have provided just represent qualitative mechanical mapping and not quantitative characterization. The higher stiffness of macerals with higher concentration of aromatic carbon has been known, but the stiffness value for different types of macerals is not known which requires quantitative measurement.”*

Response: We agree that this manuscript reports qualitative characterization of the mechanical properties of the organic matter and that correlations between aromaticity and mechanical stiffness in organic matter has been observed before. To our knowledge, this paper represents the first report of the evolution of the chemical composition of individual macerals with maturity, enabled by the AFM-IR technique which provides quantitative chemical characterization at submicron length scales.

2. *“The authors examined the evolution of OM composition with maturation. However, they examined only one type of shale: New Albany shale. Chemical composition of OM is dependent on depositional environment and the results obtained from studying the evolution of OM within one type of shale can not be generalized to others. Thus performing the same type of analysis on different samples will result in better understanding of the impact of maturation on nanoscale OM composition.”*

Response: We agree with the reviewer that natural materials are heterogeneous and it is difficult to extrapolate measurements on one system to other systems. The goal of this work is to demonstrate a proof of principle that organic matter in shale can be characterized geomechanically and geochemically at nanoscale, as well as to provide initial results describing the evolution with maturity. Repeating this analysis on samples from other locations is part of our planned future work.

3. *“The authors did not explain about the calibration of spring constant which is an essential step in nanomechanical mapping”*

Response: We added the clarification on the calibration and modified the main text.

Changes we made to the main text regarding this question:

- We added the discussion in the “Variations of geomechanical and geochemical properties among maceral groups” section: “Prior to the AFM-IR imaging, the AFM probes are carefully selected to have nearly the same contact resonance frequency when measured on a polystyrene thin film sample.”

4. *“Many findings listed in the conclusion have been obtained by traditional measurements (such as enrichment of macerals with aromatic carbon with maturation). So one of the contributions of this study would be the distribution of different macerals within OM and its evolution with maturation.”*

Response: Yes indeed, we believe this is the first work demonstrating the distribution of different macerals’ geomechanical and geochemical properties as well as their evolution with maturity.

Response to Reviewer 3

Overall Comments:

- *“The author has presented a pioneering study of shale by the AFMIR technique. They have used optical microscopy and nanoscale infrared imaging to characterize the solid organic matter into shale rock for different maturations. They have clearly demonstrated the ability of the AFM-IR technique to follow the evolution of maturation providing a new sight of the microscopic process of petroleum generation. This paper opens a new field of application for infrared nanoscale technique. In Conclusion, I recommend to accept this paper for publication with minor corrections.”*

1. *“Page3-line64. The resolution of AFMIR technique is not 100 nm. The resolution is just limited by the radius of the tip and can achieved a few nanometers (ref30).”*

Response: The resolution of the AFM-IR technique depends on the quantity that is being measured. For topography images and mechanical stiffness images, the spatial resolution is limited by the radius of the tip, and a spatial resolution on the order of nanometers is achievable, as the reviewer notes. For chemical images, on the other hand, the resolution is also controlled by the sample’s thermomechanical properties, and such fine spatial resolution is typically not achieved [3]. A typical spatial resolution of AFM-IR chemical imaging is around 100nm [4]. To our knowledge, the best spatial resolution obtained with the AFM-IR chemical imaging is around 20 nm [5].

Changes we made to the main text regarding this question:

To clarify this issue, we revised the description of the measurement resolution to specify that the spatial resolution refers to that obtained in chemical imaging: “The typical spatial resolution of the resulting chemical image is approximately 100 nm, orders of magnitude finer than the spatial resolution attainable by traditional micro-FTIR.”

2. *“Page5-line117. The authors explain that the topography shows little contrast. Do the authors could make tapping mode image to check if the same region shows the same topography? This would just confirm that the stiffness of the surface do not disturb the topography.”*

Response: Yes, it may be possible to check the topography using tapping mode. The main application of topography imaging is to identify regions of organic matter (instead of mineral

matter). In this case, we found that those zones could be identified more readily by the mechanical maps.

3. *“Page6-line119. Authors speak about low mechanical stiffness. Could they explain this part more precisely. The frequency shift of the resonance is consistent only if it is associated to absorption.”*

Response: This comments is similar to Review #1, comment #2. Unfortunately it is not yet possible to translate the reported frequencies to quantitative elastic moduli, so the mechanical arguments remain qualitative.

4. *“page6-line134. What is the idea behind quantified. Does that mean that the authors can measure the absolute stiffness of each part of the shale?”*

Response: No, as discussed previously, this technique cannot measure the absolute stiffness of each part of shale. We intended to explain that we can obtain quantitative contact resonance (CR) frequency distribution of various macerals, where the CR frequency is positively correlated with mechanical stiffness.

Changes we made to the main text regarding this question:

To avoid potential confusion regarding the ability of this technique to measure mechanical properties quantitative, we replaced the word “quantified” with “estimated”.

5. *“Page7-line151. Does the spectra obtained by AFM-IR could be compared with FTIR? Is there a way to confirm that the AFM-IR spectra are consistent?”*

Response: Yes. The spectra measured by AFM-IR are nearly identical to those measured with bulk transmission FTIR, which has been proven both theoretically and experimentally (described in references 30 and 31 in the manuscript)

Changes we made to the main text regarding this question:

To clarify this point, we added a line to the text stating “The resulting local IR absorption spectra are nearly identical to those measured with bulk transmission FTIR, providing information on chemical composition and structural characteristics. ”

6. *“Page12-line284. Does the authors has checked the heterogeneity with smaller scan like 3x3 um?”*

Response: No. We did not check the heterogeneity with smaller scan like 3x3 um² in this study. That would be interesting to investigate for future work.

7. “Page13-line287. As explained previously the resolution is not 100 nm.”

Response: See detailed reply to the first comment.

8. “Page23-line474. There is no value for absorption and frequency maps, only min and max. Even if IR mapping is in arbitrary unit, it is really useful to know if the min value correspond to zero absorption. I suggest that the author add values to both maps.

9. “Page24-line485. Same as 8).”

Response: Thank you for the suggestion. We added values to both maps.

Changes we made to the main text regarding this question:

We added the min and max value in the figure 1 and figure 2.

10. “Page27-line518. Does the spectra are taken at the same position after maturation?”

Response: No. For organic petrographic studies, shale fragments are embedded in epoxy before examination under optical microscope. The artificial maturation was performed on the same batch of New Albany shale sample, and multiple pellets are made from shale matured in different stages. The measurements are made on those pellets, not from the same position of the organic matter.

References:

1. Emmanuel, S., et al., *Impact of thermal maturation on nano-scale elastic properties of organic matter in shales*. Marine and Petroleum Geology, 2016. **70**: p. 175-184.
2. Dazzi, A. and C.B. Prater, *AFM-IR: Technology and Applications in Nanoscale Infrared Spectroscopy and Chemical Imaging*. Chemical Reviews, 2017. **117**(7): p. 5146-5173.
3. Felts, J.R., et al., *Nanometer-Scale Infrared Spectroscopy of Heterogeneous Polymer Nanostructures Fabricated by Tip-Based Nanofabrication*. ACS Nano, 2012. **6**(9): p. 8015-8021.
4. Centrone, A., *Infrared imaging and spectroscopy beyond the diffraction limit*. Annual Review of Analytical Chemistry, 2015. **8**: p. 101-126.

5. Katzenmeyer, A.M., et al., *Absorption Spectroscopy and Imaging from the Visible through Mid-Infrared with 20 nm Resolution*. *Analytical Chemistry*, 2015. **87**(6): p. 3154-3159.

REVIEWERS' COMMENTS:

Reviewer #1 (Remarks to the Author):

In the rebuttal letter the authors have satisfactorily answered my questions and in my opinion the manuscript is ready for publication.

Reviewer #2 (Remarks to the Author):

I recommend the manuscript for publication. However, the qualitative nature of mechanical characterization needs to be clarified in the manuscript.

Reviewer #3 (Remarks to the Author):

The corrections that I have suggested have been made. I propose that the paper to be published.

Response to Reviewers

The reviewers had one remaining comment on the manuscript: “However, the qualitative nature of mechanical characterization needs to be clarified in the manuscript” from the second reviewer. In response, we added the following discussion to the manuscript: “quantitatively translating the measured CR frequency to stiffness requires extensive modeling of the mechanical response of the cantilever to the photothermal expansion of the sample, so stiffness in AFM-IR measurements is typically described using the CR frequency²⁸⁻³⁰”.